# All-in-One: Boosting Basic Capabilities in one Omni-MLLM to Enhance Movie Understanding

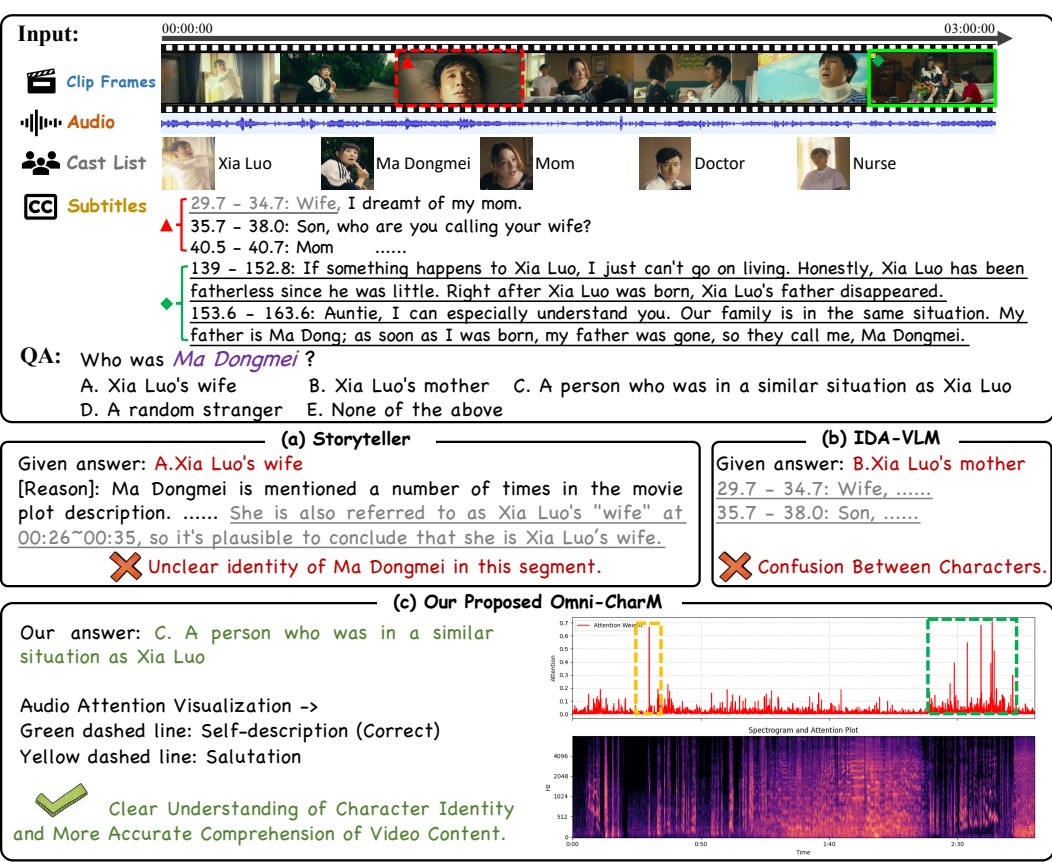

Figure 1: Illustration of how Omni-CharM works and comparison with existing methods.

## Abstract

Movie understanding is still challenging as a movie involves many characters with complex relationships and it is edited for appealing audiences, which are neglected in current multimodal large language models (MLLMs). Only a few previous works propose ideas to identify characters and integrate ID information in models, but they use cascaded models, or vision and scripts only while ignoring audio. To address these problems, we propose an all-in-one Omni-MLLM with built-in basic capabilities of ID identification, shot-level description, and critical sub-question answer in thinking. First, we construct identity related data consisting of 12 fine-grained character-centric tasks to improve the model's ability to identify characters from vision and audio. Second, we leverage frame and shot descriptions to alleviate the difficulty of training. Third, we explore how to enhance our model further by using Chain of Thought (CoT) data from an advanced model. Experimental results show that our proposed model achieves stable improvements on both ID-Aware Movies Understanding questions' set StoryQA and general video understanding benchmark VideoMME. Ablation studies confirm the positive contributions from all of our proposed ideas.

# 1 INTRODUCTION

The rapid advancement of multimodal large language models (MLLMs) Xu et al. (2025) Comanici et al. (2025) Hurst et al. (2024) has demonstrated unprecedented capabilities in understanding complex multimodal information. However, as an art form that integrates visual, auditory and textual information, cinema remains challenging for MLLMs to comprehend. Recent research works improve model's movie understanding ability by adopting a phased strategy and training model on specific tasks like instance understanding in short videos. StoryTeller He et al. (2024) employs a phased methodology by training an Omni-MLLM on speaker naming task and a MLLM on dense description task respectively in different stages, while such framework is time-consuming, error-accumulating and model in each stage is restricted to the specific task. Other works Peng et al. (2024); Ji et al. (2024) improve model's ID recognition ability via instruction tuning, while the models only support visual and text modalities, and is restricted to the seconds-level task. Nevertheless, character identity and development are often established over long temporal spans, demanding strong long-term memory and complex reasoning across modalities.

To address this challenge, we propose an all-in-one Omni-MLLM with enhanced basic abilities of character identification, shot-level description and critical sub-question answer in thinking. We employ an end-to-end multiple-choice evaluation strategy on character-level movie comprehension benchmark *StoryQA* by directly feeding minutes-long movie clip (frames, audio and subtitles excluding speaker), cast list (character faces and names) and QA into the model. As shown in Figure 1 (a), StoryTeller fails to answer correctly due to the misleading of speaker at 00:26-00:35, once stage two's task fails, stage three's MLLM is hard to bridge the gap of losing audio input. While IDA-VLM is also misled by the subtitles 'Wife' (shown in Figure 1 (b)), our model is capable of answering the question by integrating audio and visual information. We analyze our model's audio attention results. As shown in Figure 1 (c), unlike other models that focus on distractors (yellow dashed line, corresponding to the input video/subtitles red dashed line), our model concentrates on the final audio segment (green dashed line, corresponding to the input video/subtitles green dashed line), demonstrating stronger comprehension and producing the correct answer.

Our model is developed from a dual-stage pipeline consisting of Basic Boosting Stage and Task Adaptability Stage. Since there is no high-quality omni-modality data as suitable for boosting model's character-level movie comprehension capabilities, in terms of eliciting and assessing reasoning ability, we begin by meticulously crafting omni-modality ID-related Data, Frame and Shot Description Data, and task adaptation data MovieQA by an automated data generation pipeline based on annotations in MovieStory101 He et al. (2024), movies from Movie101 Yue et al. (2023) and Movie101v2 Yue et al. (2024), to empower model with basic ID-Aware capabilities. These tuning data unleash the MLLMs' built-in ID-Aware capabilities, shot-level and frame-level Description ability, which contributes to our model performance on *StoryQA*.

Furthermore, inspired by DeepSeek Guo et al. (2025)'s success on Chain-of-Thought(CoT) training, we create a CoT Data based on MovieQAby using advanced model Gemini2.5-Pro Comanici et al. (2025). We perform the instruction tuning to our model on CoT Data and conduct systematic experiments on the influence of CoT prompting when evaluating model on the benchmark. We identify an unusual insight for researchers that would like to dive deeper into arousing MLLMs' reasoning ability on movie understanding, that is, omni-MLLMs can not diretly learn complex CoT ability from advanced model by simply training it on a small amount of reasoning data, while the reasoning data may potentially unleash model's performance when infer model without CoT. To this end, our Basic Boosting Stageinvolves training our model on ID-related Data, Frame and Shot Description Data and CoT Data, while Task Adaptability Stage involves training on MovieQA. We also perform comprehensive ablation experiments on the composition of the Basic Boosting Stage. Results demonstrate the effectiveness of our every data component and training pipeline.

Comprehension of cinematic characters demands a more intricate and cascading cognitive process than that required for discrete tasks. The demand for the integration of character audio and video with MLLM is significant. In this paper, we methodically and thoroughly examine the impact of diverse character-related tasks under the supervised fine-tuning (SFT) on various responses to character-related movie comprehension inquiries. Our primary focus is on enhancing the character-level movie comprehension of existing MLLMs, with the objective of providing valuable insights for future research. The following conclusions are the primary results of this study:

- We introduce high-quality omni-modality ID-related Data of 12 tasks, Frame and Shot Description Data and CoT Data for boosting model's ID-Aware capabilities.

- We introduce an Omni-CharM model, a model trained on dual-stage pipeline:Basic Boosting Stage and Task Adaptability Stage, enhancing basic built-in ID recognition, shot-level description ability towards Movie Understanding.

- Extensive experiments demonstrates that our Omni-CharM achieve stable improvements across ID-Aware movie understanding questions'set StoryQA and general video benchmark Video-MME. Each task in our Basic Boosting Stage has been proven efficient.

- We discover that open-source Omni-MLLM can't directly learn complex reasoning pattern by simply fine-tuning on CoT Data from advanced closed-source model. However, model after fine-tuning would gain an improvement when infer without CoT.

## 2 RELATED WORK

### 2.1 IDENTIFICATION-AWARE MULTIMODAL UNDERSTANDING

Character identification is a fundamental challenge in movie understanding, requiring models to recognize and track individuals across dynamic scenes. Traditional approaches relied on face detection and re-identification techniques, but these methods often struggled with occlusions, lighting variations, and non-frontal poses Cheng et al. (2020) He & Liu (2020) Zhao et al. (2003). Recent advances in Multimodal Large Language Models (MLLMs) have improved character disambiguation by integrating visual, textual, and contextual cues. For instance, StoryTeller He et al. (2024) leverages audiovisual dialogue matching to link characters to their names, yet it operates in a pipeline manner, separating identification from description generation. Similarly, IDA-VLM Ji et al. (2024) highlights the limitations of existing Large Vision Language Model in cross-scenario identity association and proposes instruction tuning with ID references, but it primarily focuses on visual-instance matching without end-to-end audiovisual-textual fusion. In contrast, our method adopts an end-to-end framework that jointly optimizes character identification and narrative reasoning by synchronously processing visual appearances, speech, and dialogue context. Experiments in Section 4 indicate that our approach ensures effective identity aware understanding of movies, while maintaining strong abilities of general video understanding.

### 2.2 SHOT-AWARE MULTIMODAL UNDERSTANDING

Film and television productions contain rich shot-level information. Directors employ diverse shot techniques to convey narrative cues, emotions, and pacing. Recently, an increasing number of works have explored multimodal film understanding by analyzing shots Han et al. (2025); Xie et al. (2025); Liu et al. (2025); Gu et al. (2023). Xie et al. (2025) introduced Shot-by-Shot, a training-free audio description framework that uses temporal context based on shots, thread structure, and scale-dependent prompts to provide film grammar cues. This framework achieved state-of-the-art performance on CMD-AD, TV-AD, and MAD-Eval, surpassing previous training-free and fine-tuned baselines. Similarly, Liu et al. (2025) propose MovieChat-2, a MLLM that fuses shot-level features with text and audio to improve temporal reasoning and narrative comprehension for long-form videos, outperforming prior methods on MovieQA and MovieNet. Inspired by them, we propose using shot descriptions as a complement to frame descriptions during training and achieve better performance in movie understanding task in this paper.

### 2.3 CHAIN OF THOUGHT IN MULTIMODAL REASONING

Researchers have discovered that Chain-of-Thought (CoT) reasoning, a technique that encourages large language models (LLMs) to simulate human-like reasoning step by step when solving complex problems, can significantly improve performance on arithmetic, commonsense, and symbolic reasoning tasks by decomposing complex problems into more manageable sub-problems Kojima et al. (2022) Zhan et al. (2025) Wei et al. (2022). The success of CoT in the text domain has inspired researchers to extend this paradigm to the multimodal realm, giving rise to Multimodal Chain-of-Thought (MCoT) reasoning, e.g., Multimodal-CoT Zhang et al. (2023) demonstrates incorporating multimodal information into a dual-stage CoT framework can efficiently improve model's multimodal

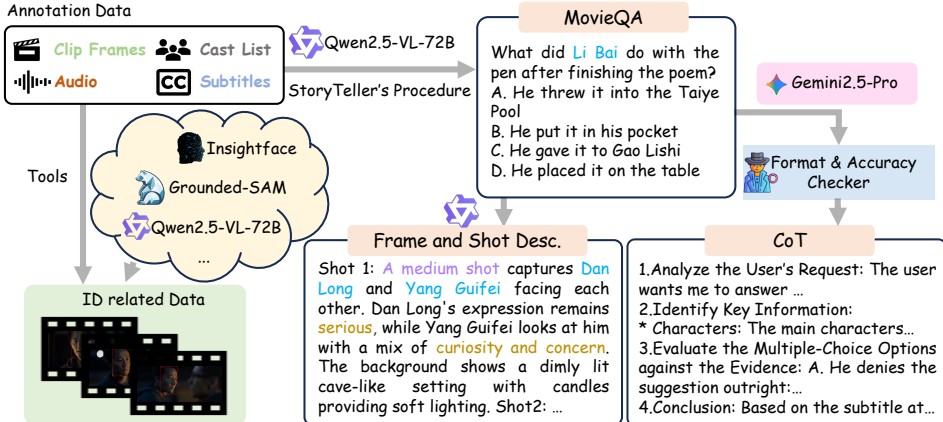

Figure 2: Data construction pipeline. Based on metadata containing clip frames, cast lists, audio, and subtitles, we construct various types of omni-modality data.

commonsense reasoning. In the context of video understanding, Video-R1 Feng et al. (2025) success by eliciting MLLM's temporal modeling ability for video reasoning by leveraging CoT training. Video-SALMONN-o1 Sun et al. (2025) improves LLM's audio-visual reasoning ability across different video reasoning benchmarks. However, audio-visual reasoning under character-related movie understanding still remain unexplored. To the best of our knowledge, Omni-CharM first explores how CoT can benefit Omni-MLLM's ID-related movie understanding ability.

## 3 METHODOLOGY

### 3.1 PROBLEM DEFINITION AND FRAMEWORK OVERVIEW

This paper aims to address the problem of ID-aware movie understanding. As the Figure 1 shows, given a three-minute movie clip (including frames, audios and subtitles but without the information who is the speaker), a cast list (including character photos and corresponding names) and a character-related question, the model must integrate information from omni-modalities to generate an answer from multiple choices for a question.

To address this this issue, we propose an all-in-one Omni-MLLM called Omni-CharM based on the model Qwen2.5-Omni-7B Xu et al. (2025) developed from our dual-stage training pipeline via supervised fine-tuning (SFT). (1) *Basic Boosting Stage* involves the mix training of ID-related tasks, which enhances the model's ability to align character IDs with their faces or speech, adding frame and shot descriptions, which makes training easier to converge, and using CoT data for training, which enhances the model's ability to answer some sub-questions in CoT. Thus basic abilities are boosted to answer identity related movie understanding questions. (2) *Task Adaptability Stage* adapts the model to the task of answering ID-aware movie understanding questions in specific formats. To facilitate the Basic Boosting stage, we also propose a data annotation pipeline to construct rich fine-tuning data.

### 3.2 DATA ANNOTATION PIPELINE.

We propose an automated data construction pipeline designed to generate abundant data sourced from video. As shown in Figure 2, initially, based on the annotation data from MovieStory101 He et al. (2024) and movies from movie101 Yue et al. (2023) and movie101v2 Yue et al. (2024), we utilize advanced tools like Grounded-SAM Ren et al. (2024) for segmenting character and acquire bounding-boxes, insightface Cheng et al. (2020) for recognizing characters from different images, to acquire our ID-related Data. Then to adapt our model with our task, we follow StoryTeller He et al. (2024)'s procedure of constructing StoryQA and construct MovieQA on our training set by using Qwen2.5-VL-72B Bai et al. (2025). Since character-level understanding is not limited to recognize characters, inspired by Xie et al. (2025), we further introduce Frame&Shot description data and chain-of-thought data respectively by using Qwen2.5-VL-72B and Gemini2.5-Pro Comanici et al.

Table 1: ID-related data statistics and task description. ( **A.** indicates Audio; **V.** indicates Visual; **T.** indicates Text; **C.** indicates Cast list; **S.** indicates Subtitles; ASR indicates Audio Speech Recognition; Desc. indicates Description; Recog. indicates Reocgnition.**Input** indicates Input Modality)

| Task Name | #Sample | Input | | | | | Desciption |
|---|---|---|---|---|---|---|---|
| | | A. | V. | T. | C. | S. | |
| Robust ASR | 5,017 | ✓ | | ✓ | | | Recognize speech from audio derived from movies. |
| Audio Diarization | 8,476 | ✓ | | ✓ | | | Partition an audio into segments based on speaker identity. |
| Audio Recog. | 8,787 | ✓ | | ✓ | | | Recognize audios from the same speaker. |
| Character Image Desc. | 1,791 | | ✓ | ✓ | ✓ | | Describe image with ID from cast list. |
| Face Localization | 12,581 | | ✓ | ✓ | | | Output faces' bboxes in the image. |
| Character Tracking | 32,363 | | ✓ | ✓ | ✓ | | Output recognized ID and bboxes in frames. |
| Character Recog. | 46,418 | | ✓ | ✓ | ✓ | | Output recognized ID and bboxes in the image. |
| Character Video Desc. | 1,260 | ✓ | ✓ | ✓ | ✓ | ✓ | Describe the events in the video with ID. |
| Character Appearance Desc. | 1,671 | ✓ | ✓ | ✓ | ✓ | ✓ | Recognize characters who appear in the given frames and describe their appearance. |
| Active Speaker Localization | 2,171 | ✓ | ✓ | ✓ | ✓ | ✓ | Output recognized speaker and bbox in each frame. |
| Character Emotion Recog. | 3,272 | ✓ | ✓ | ✓ | ✓ | ✓ | Recognize speaker's ID and tell his emotion from choices given. |
| Speaker Naming | 5,176 | ✓ | ✓ | ✓ | ✓ | ✓ | Output identified speaker and his speaking time. |

(2025) to investigate their role in enhancing model's ID-aware movie understanding ability. Detailed data construction process is provided in Appendix A.

### 3.3 BASIC BOOSTING STAGE

Mirroring human learning processes, basic capabilities in recognizing characters in essential for Omni-MLLM to learn. To this end, we first train our model on our meticulously designed omni-modality ID-related Data of 12 tasks, Frame and Shot Description Data , and chain-of-thoughts data.

**Boosting ID-aware Capabilities.** Movies often feature complex changes in lighting and shadow, as well as shifts in character poses. This makes it necessary to integrate multimodal information in order to recognize characters. As shown in Table 1, unlike traditional datasets such as AISHELL-4 Fu et al. (2021) or AMI Kraaij et al. (2005), which primarily focus on single modality, our ID-related Data  spans over both different temporal and omni-modalities character-related scenarios from movies, enabling models to better utilize cross-modal information and more adaptable to the diverse and complex contexts in which characters appear. Additionally, our ID-related Data is more closely mirrors the way humans associate voices, faces, and contextual cues to recognize and remember characters. This richer coverage not only improves speaker identification but also strengthens a model's ability to answer character-related questions in broader movie understanding scenarios.

**Frame and Shot Descriptions.** Inspired by filmmaking practices, we incorporated shot scale information into the character speaker task, emphasizing the character of shot scale. To this end, we constructed Frame and Shot Description Data. By setting prompts, we guide the model to analyze the scale type (five types), thereby focusing on the understanding of the image at different levels (such as focusing more on the environment or more on details). *(1) Frame-level Scale Annotation.* Given a list of video frames and a list of characters (including their names), we employ Qwen2.5-VL-72B-Instruct Bai et al. (2025) model to generate content descriptions for each frame. Each description should explicitly mention character names and prioritize details according to the scale type of the current frame: For *Close-up* or *Extreme Close-up* shots, will focus on the characters' facial expressions and emotional states. For *Medium* shots, emphasis should be placed on the interactions

Table 2: Main results on StoryQA and VideoMME benchmark. (**Bold** and underline indicate the best and the second-best results. **Char.** indicates character. **S.** indicates short, **M.** indicates medium and **L.** indicates long.)

| Method | ID-Aware | StoryQA | | | | Video-MME | | | |
|---|---|---|---|---|---|---|---|---|---|
| | | Char. | Action | Plot | Total | S. | M. | L. | Avg. |
| *Closed-Source Baselines.* | | | | | | | | | |
| Gemini2.5-Pro | ✗ | **74.66** | 72.30 | 78.37 | 74.65 | **81.7** | **69.0** | **64.1** | **71.6** |
| Gemini2.5-Flash | ✗ | 72.09 | 68.64 | 72.79 | 70.63 | 74.1 | 52.6 | 46.7 | 57.8 |
| *Open-Source Baselines* | | | | | | | | | |
| AVicuna | ✗ | 29.56 | 27.29 | 30.01 | 28.91 | 20.6 | 23.2 | 20.0 | 21.3 |
| IDA-VLM | ✓ | 41.15 | 41.40 | 40.39 | 41.08 | 52.2 | 39.7 | 35.0 | 42.3 |
| VideoLLaMA2 | ✗ | 54.43 | 51.95 | 53.38 | 53.41 | 60.6 | 47.7 | 41.1 | 49.8 |
| VITA1.5 | ✗ | 64.48 | 66.56 | 70.05 | 66.30 | 66.2 | 54.3 | 49.4 | 56.7 |
| StoryTeller | ✓ | 67.60 | 58.30 | 64.40 | 63.90 | 47.7 | 39.4 | 34.7 | 40.6 |
| Qwen2.5-Omni-7B | ✗ | 65.49 | 65.40 | 70.26 | 66.43 | 71.0 | 61.1 | 52.8 | 61.6 |
| **Omni-CharM** | ✓ | 74.36 | **76.44** | **80.72** | **76.33** | 72.9 | 62.0 | 52.7 | 62.5 |
| *Imp. over Qwen.* | - | +13.5% | +16.8% | +14.9% | +14.9% | +2.7% | +1.5% | -0.2% | +1.5% |
| w/o BasicBoosting | ✓ | 73.13 | 75.42 | 78.38 | 74.95 | 71.7 | 62.0 | 53.3 | 62.3 |
| w/o TaskAdaptation | ✓ | 74.33 | 76.22 | 80.22 | 76.15 | 72.6 | 62.1 | 52.0 | 62.2 |

between characters and their body movements. For *Long* or *Full* shots, priority should be given to the environmental context, including spatial composition, positioning, and overall atmosphere. *(2) Shot-level Scale Annotation.* Given a video and a list of characters (with names), we first use the same MLLM model to segment the video into individual shots and assign each shot a unique identifier. For each shot, we then generate a description containing character scale information, following the same focus criteria as in the frame-level annotation.

**Chain-of-Thought for Training.** Inspired by DeepSeek Guo et al. (2025) and considering the complexity of the evaluation task, we decided to explore whether reasoning chains could further enhance the model's performance. Specifically, we use MovieQA obtained from the training set and emulate the final evaluation process of the model by inputting it into Gemini 2.5-Pro Comanici et al. (2025) to generate a series of answers and chain of thought data. Following the format and answer filtration process, the CoT data was constructed, encompassing responses that adhered to the prescribed format, along with the underlying chain of thought that substantiated their accuracy. As shown in Figure 6, data derived from Gemini2.5 Pro Comanici et al. (2025) exhibits reasonable CoT Process.

## 3.4 TASK ADAPTABILITY STAGE

In order adapt the model to answering QA and to explore the impact of QA on improving the model's ability to answer character-related questions, we modeled StoryQA after StoryTeller's construction of StoryQA, and use Qwen2.5-VL-72B-Instruct Bai et al. (2025) to generate the data MovieQA on MovieStory101's training set.

## 4 EXPERIMENT

In this section, we conduct experiments to compare our proposed method with state-of-the-art baselines and ablation studies to analyze the contributions from different proposed ideas.

## 4.1 EXPERIMENTAL SETUP

**Benchmarks and Evaluation Metrics.** We evaluate methods on two benchmarks to validate whether it improve the performance on ID-aware movie understanding without sacrifice of performance on general video understanding:**(1) StoryQA** He et al. (2024) is a movie comprehension benchmark with character IDs generated from manually annotated movie descriptions. **(2) Video-MME** Fu et al. (2025a) is a widely-used comprehensive evaluation benchmark of Omni-MLLM in

Table 3: Ablation studies on SFT data of Basic Boosting Stage. (Char. Related indicates Character Related Data; F & S Desc. indicates Frame & Shot Description.) There are two modes to apply CoT: 1) using CoT for training only and 2) using CoT for training and testing. We find that our model may generate errors when inferring chain-of-thoughts and thus enabling CoT in testing harm the performance; whereas, using CoT for training only cna enhance some basic capabilities of our Omni-CharM .

| Basic Boosting Stage | | | StoryQA | | | |
|---|---|---|---|---|---|---|
| Char. Related | F & S Desc. | CoT | Character | Action | Plot | Total |
| | | | 73.13 | 75.42 | 78.38 | 74.95 |
| ✓ | | | 74.36 | 75.20 | 78.81 | 75.54 |
| | ✓ | | 73.80 | 75.55 | 79.42 | 75.53 |
| | | ✓(train only) | 73.81 | 75.51 | 78.38 | 75.29 |
| | | ✓(train&test) | 64.09 | 65.18 | 66.87 | 65.01 |
| ✓ | ✓ | | 73.96 | 76.00 | 80.08 | 75.87 |
| ✓ | | ✓ | 74.33 | 75.51 | 78.88 | 75.64 |
| | ✓ | ✓ | **74.79** | 76.22 | 79.80 | 76.28 |
| ✓ | ✓ | ✓ (train only) | 74.36 | **76.44** | **80.72** | **76.33** |
| ✓ | ✓ | ✓(train&test) | 57.58 | 60.59 | 62.57 | 59.58 |

video analysis, in which people usually have no name. As all questions have options to choose, we can use **Accuracy** of how many chosen answers are correct to measure results.

**Compared Methods.** We compare our model with three types of baselines. **(1)ID-Aware Methods.** There are two baselines proposed to solve ID-aware movie understanding problems: IDA-VLM Ji et al. (2024) is an end-to-end model and we follow their inference instruction and crop cast list into single image to provide cross-attention on characters; Storyteller He et al. (2024) is three-stage pipeline method and we report their scores from paper on the same dataset. **(2) General Methods.** We include several high-performing general Omni-MLLMs as general baselines: AVicuna Tang et al. (2024), VideoLLaMA2 Cheng et al. (2024), VITA1.5 Fu et al. (2025b), Qwen2.5-Omni-7B Xu et al. (2025) and Gemini-2.5-Pro Comanici et al. (2025). For those models, we perform prompt engineering to generate corresponding QA answer. The prompt examples are provided in Appendix. **(3) Our Methods.** We take Qwen2.5-Omni-7B as our backbone and apply two-stage fine-tuning, i.e., one for boosting basic capabilities and the other for adapting to the StoryQA task, to get our final Omni-CharM model. We also report two ablations of Omni-CharM without one of the two stages: Omni-CharM w/o BasicBoosted and Omni-CharM w/o TaskAdapated.

**Implementation Details.** We train our models based on Qwen2.5-Omni-7B Xu et al. (2025) via supervised fine-tuning for one epoch using LLaMA-Factory framework Zheng et al. (2024). 4 NVIDIA H100 80G GPUs are used to train and test the models. For open-source models, we apply 0.25 frames per second(fps) to handle the video clip. For closed-source models, we use the default 1 fps as evaluation setting.

## 4.2 MAIN RESULTS

We evaluate our Omni-CharM and the baselines on both StoryQA and Video-MME datasets and present results in Table 2. Our proposed Omni-CharM outperforms all open-source baselines on both datasets. Notably, Omni-CharM achieves a significant improvement of 14.9% over its backbone Qwen2.5-Omni-7B and even 2.5% over the strongest close-source baseline Gemini2.5-Pro on our target StoryQA dataset. This indicates that our proposed method is effective to improve complex movie understanding involving IDs. At the same time, our Omni-CharM performs slightly better than its Qwen backbone. It indicates that our proposed method has good generalization ability. In contrast, the ID-aware movie understanding baseline StoryTeller is ranked lower in terms of its performance on Video-MME than that on StoryQA, i.e., Top 4 vs. Top 6 among open-source methods. And another ID-aware MLLM baseline IDA-VLM performs worse on both datasets.

## 4.3 Ablation Studies

**Effect of Basic Boosting Tasks.** To investigate the effectiveness of each component of basic boosting stage, We conducted an extensive ablation study on various configurations and present results in Table 3. First, we use only one proposed component, i.e., character related tasks, frame&shot descriptions, or chain-of-thoughts. The results indicate that every module has positive contributions. The character related tasks contribute most to the questions of character category; whereas, the frame&shot descriptions contribute most to the questions of plot category. These make sense as the fune-tuning tasks are highly relevant to the test questions. Second, the combination of any two components can further boost the model's performance, surpassing the performance achieved when each is used as a standalone. The result suggests that we can achieve the most improvement when combining frame&shot descriptions and CoT data for training only, which can also perform better for the questions of character category. This indicates that the two component are complementary and can cover some functions of character related tasks. Third, the model based on all components achieves the best overall result, in particular for the questions of plot category. This indicates that the plot questions may need more complex reasoning according to character and plot.

**Effect of Individual Character Related Task.** To better investigate the effect of 12 individual character related tasks to ID-aware movie comprehension, we do an ablation study by removing the training data of each task. All results are presented in Figure 3. Specifically, to save time, we first randomly sample 300 pieces of data from each character related task. Then we fine-tune the backbone model by using all $12 * 300$ samples in the BasicBoosted stage and all QA data in the TaskAdapted stage, we denote it as *FullCharacter* (See the brown dotted circle in Figure 3). Also we choose the model Omni-CharM w/o BasicBoosted as the baseline *w/o AnyCharacter* (See the black dotted circle in Figure 3). Next, we remove 300 samples from each character related task one by one and train ablation models without one character task (See the brown dots on different spokes of the figure).

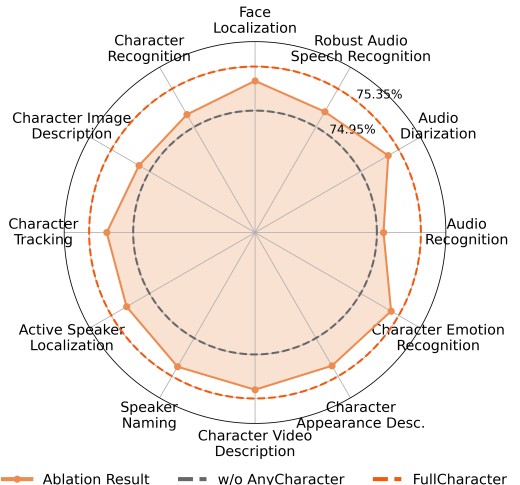

Figure 3: Ablation study on character data types, showing that all sub-tasks improve performance, with the *FullCharacter* setting performing best (brown dotted circle).

We have some interesting findings from the results. First, all results of removing any character related task are between the performance of *FullCharacter* and *w/o AnyCharacter*. This indicates that all tasks have positive effect to the best performance. Second, the most useful task is audio recognition perhaps because it provide speech content that is valuable to comprehension. Third, the top 2 and 3 useful tasks are character recognition and character image description, which have direct relationship to recognize who he/she is from image. In summary, all 12 tasks are necessary and useful.

**Effect of Shot and/or Frame Descriptions.** We do an ablation study to show the effectiveness of adding frame and/or shot descriptions that are generated by Qwen2.5-VL-72B Bai et al. (2025) in training phase. As shown in Figure 4a, adding both frame and shot descriptions performs better than adding frame or shot description, which brings gains over the baseline without frame and shot descriptions too.

To investigate why this happens, we draw the training curves of each model. As shown in Figure 4b, the more description added the lower the loss is achieved and faster. This may indicate that although we have no such description provided during testing, such descriptions alleviate the difficulty of converge during training as they make the model easier to generate right answers. More important, without description in testing, the model still works better, which demonstrates that the model can learn better relations between the answer and the other parts of input, rather than the descriptions.

Furthermore, we visualize attention from the generated answer to vision patches in key frames. As the example in Figure 5 shows, we find that with frame and shot descriptions provided in training,

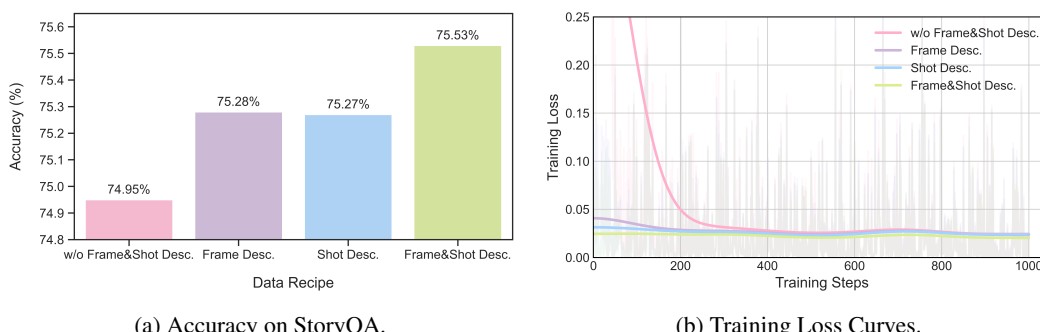

(a) Accuracy on StoryQA.         (b) Training Loss Curves.

Figure 4: Impact of adding different descriptions in training. (a) Adding both frame and shot descriptions performs the best in testing. (b) Adding frame and shot description in training can speed up converge and obtain lower loss in training

the model can build stronger attention to the character's face, in particular her mouth, and the object mentioned in the question, such as "mirror". That may be another reason why such a component works.

**Effect of CoT for Training Only.** It is a surprise finding that CoT data is better used for training only in our experiments. As shown in Table 3, when we infer with CoT turned on, model performance drops significantly compared to inferring without CoT. Directly training models with small amounts of complex reasoning data from advanced closed-source models fails to stimulate their multimodal reasoning capabilities. It's akin to teaching a struggling

What did *Zhao Bing* do first when *she* was in front of the mirror?
A. She combed her hair.  B. She splashed water on her face.
C. She picked up a towel.  D. She put on makeup.

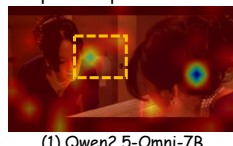 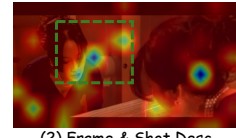

(1) Qwen2.5-Omni-7B   (2) Frame & Shot Desc.

Figure 5: Visual attention of model results: ours model stronger focus on the face in the mirror, showing better understanding.

student by simply giving them the a few top student's chain-of-thoughts. After training, although the struggling student may learn the steps but often fails when she/he gives wrong answer to some step. In contrast, if the struggling student adopts their own approach, that is, answering questions directly without applying CoT after training, they may actually show improvement perhaps due to their enhanced capabilities of solving some sub-problems and generated confident procedure. As shown in Figure 6, our model only learns the template of CoT from advanced model but generates something wrong during the long chain of thoughts.

### 4.4 LIMITATIONS AND DISCUSSIONS

The ID-related Data generation process involves multiple models, such as sentiment recognition models. The current accuracy of these models needs improvement, which necessitates enhanced data quality. In essence, models require exposure to diverse data to develop foundational capabilities and steadily enhance their reasoning abilities. We believe that future improvements in data quality will lead to better model performance.

## 5 CONCLUSION

In this paper, we introduce an all-in-one model Omni-CharM with built-in basic capabilities of ID identification, frame and shot description, and critical sub-question answer in thinking. First, in the Basic Boosting Stage, we construct 12 fine-grained tasks of data to enable the model the ability to identify characters from vision and audio. Then, we leverage frame and shot descriptions to alleviate the difficulty of training. Second, in the Task Adaptability Stage, to enhancing our model QA ability towards Movie Understanding. Third, we explore how to enhance our model further using CoT data from a strong model. Experimental results show that our proposed model achieves stable improvements on both ID-Aware Movies Understanding questions' set StoryQA and general video understanding benchmark VideoMME. Ablation studies confirm the positive contributions from all of our proposed ideas.

## ETHICS STATEMENT

We acknowledge that movie understanding technology, like other generative technologies, carries potential risks of misuse. The primary motivation for our research, however, is positive. We believe this technology holds significant potential for beneficial applications. We are committed to the responsible advancement of this field and encourage continued research into synthetic content detection and the establishment of clear ethical guidelines for deployment.

## REPRODUCIBILITY STATEMENT

To ensure the reproducibility of our work, detailed experimental information can be found in Appendix. In the future we will also open source all the training code and more details about prompt construction data. We are committed to transparency and facilitating future research in this area.

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

## A  ID-RELATED DATA CONSTRUCTION DETAILS

**Audio Tasks.**  For audio-related tasks, we designed audio diarization, audio recognition and robust audio speech recognition tasks, which aim to improve the model's ability to extract character audio and tones from complex scenes in movies, and then better correspond the character names to the tones.

- **Audio Diarization.** For audio diarization task, given audio file consisting of WAV audio segments from different speakers, each ranging from 5 to 30 seconds, the model needs to distinguish between speakers in different time periods and output them in the format of [start,end]:speaker_id. The data was created by iteratively sampling shorter segments from the original audio clips in MovieStory101 He et al. (2024) dataset, then re-evaluating speaker presence to include only intervals longer than 0.3 seconds, and ensuring that each segment contained between 3 and 10 different speakers. Timestamps for speaker activity were adjusted relative to the start of these new segments, and the final output for each segment includes unique speaker identifiers and their sorted start and end times.

- **Audio Recognition.** For the audio recognition task, the model's goal is to identify which of four provided audio options is spoken by the same person as a given query audio utterance, where the answer is the label (e.g., A, B, C, D, or E for "None of the above") corresponding to the correct multiple choice option. The data was created by first organising the speaker utterances pre-segmented by the diarisation data provided by StoryTeller He et al. (2024), retaining those longer than two seconds, and creating a dictionary for each film where the key is the speaker name and the value is the corresponding audio clip path. For each speaker with at least two audio clips, one utterance from a target speaker was designated as the query audio and another different utterance from the same speaker served as the correct answer, while three distractor utterances from different speakers within the same original clip were selected; these four audio options were then mixed to form the question.

- **Robust Audio Speech Recognition.** For the Automatic Speech Recognition (ASR) task, model is asked to transcribe spoken audio utterances into text, with the expected output being the ground truth transcription for a given audio segment. This task was created by randomly sampling approximately 5% of utterances from the diarization data provided by StoryTeller He et al. (2024), for which transcriptions and timing information were already available. For each selected utterance that had a corresponding audio file, an entry was formed that included the audio file path, a human-readable prompt instructing to perform ASR, and the known ground truth transcription as the target response.

**Visual Tasks.**  For the visual task, we designed a face localisation, character recognition, character tracking and character image description task that moves from image to video and from global to local. Since in the process of movie understanding, we will confirm the identity of the character from the perspective of the character's face and appearance, so to make the model have the ability to recognize the location of the character and describe the character from the scene in the movie, given the character image and a scene in the movie, we constructed the following four tasks.

- **Face Localization** For the face localization task, model is required to identify and outline facial regions within image frame, with the expected output for each frame being a sorted list of normalized bounding boxes. The data was created by randomly selecting frames from video clips, performing facial detection on them by calling the buffalo_l Model, and then adjusting the detected bounding boxes to align with Qwen2.5-Omni-7B's image pre-processing step where the original image's shortest edge was resized to 448 pixels, followed by a 448x448 center crop. After this adjustment, the bounding box coordinates were normalized, filtered to remove boxes that were too small or outside the cropped area, and finally rounded to two decimal places and sorted.

- **Character Image Description.** For the character image description task, model is supposed to generate detailed textual descriptions of image scenes, with a specific focus on identifying pre-defined cast characters by name within those scenes; the expected output is an id-aware textual description, which explicitly names at least one of the known cast characters depicted. The data was created by providing a Qwen2.5-VL-72B Bai et al. (2025) with selected image frame, along with names and cropped facial images for each character in the associated cast

list. The model was instructed to describe the scene and identify any cast characters present by name, and its generated textual descriptions were subsequently filtered to include only those that were non-empty and mentioned at least one cast member.

- **Character Recognition.** For the character recognition task, given a cast list, characters' bounding boxes and name in the cast list and a video frame, model needs to identify known characters within video frame and providing their locations; the expected output for a processed frame is a list of dictionaries, each mapping a recognized character's name to their normalized [0,1] bounding box coordinates, adjusted relative to a standard 448x448 cropped view. This data was created by first building a face library of character embeddings from cast list images for each video clip. Subsequently, frames were randomly sampled from these clips, and faces detected within them by the buffalo_l Model were matched against the established library by calculating cosine similarity; for recognized characters, their bounding boxes were processed and filtered the same as face localization, with the resulting character names and their adjusted bounding boxes forming the target output.

- **Character Tracking.** For the character tracking task, given multiple video frames and known characters, the model must identify a character and track him across multiple video frames and output his location in each frame; the expected output for each instance is a structured string representing a dictionary that maps the character's name to a list of its bounding boxes for the frames within that particular chunk. This data was created by first aggregating the per-frame character recognition results for each film, detailing where each character appeared and its corresponding bounding boxes. The characters that appeared at least twice in each film and were recognised at least twice were then used to construct a tracking data set with less than 10 frames. Each data entry was then formed by combining the frame paths for one such chunk, sample cast images of the character, and a prompt, with the target output being the character's name mapped to their list of bounding boxes for those specific frames.

**Audio and Video related Tasks.** In the context of audio and video-related tasks, the following activities were designed: active speaker localization, speaker naming, character video description and video character appearance description. The purpose of this model is to fuse audio and video information in order to derive the current character ID. This is then used to facilitate more complex inference tasks.

- **Character Video Description.** For the character video description, we extend character image description task to video level. Given a short video clip, subtitles without speakers, cast list and known characters, model is required to generate character-aware textual descriptions of short video segments, requiring the identification of known cast characters by name; the expected output is a non-empty, model-generated textual description of the video segment that successfully identifies and names any present cast characters. The data was created by providing Qwen2.5-VL-72B Bai et al. (2025) with multi-modal inputs for each pre-defined video segment, including the video file itself, its audio, several extracted frames, relevant subtitles, and detailed cast information (character names and their facial images). The model was instructed to generate a comprehensive description, specifically naming any cast characters it identified, and the resulting textual descriptions were subsequently filtered to retain only non-empty outputs that successfully incorporated this character information.

- **Character Appearance Description.** For the video character appearance description, model needs to identify known cast characters within short video clips and generating structured textual descriptions of each identified character's appearance or actions. The output should be a dict mapping each recognized character's name to their respective description for that segment. The data was created by supplying a Qwen2.5-VL-72B Bai et al. (2025) with these video segments along with associated cast information, including character names and their facial images. The model was specifically instructed to identify which characters appeared in each segment and to provide descriptions of their appearance or actions in a JSON string format, and these non-empty JSON responses were then collected as the core of the data.

- **Speaker Naming.** For the speaker naming task, we construct both scene-level and clip level data. Model needs to perform speaker identification within video scenes, aiming to determine who is speaking during specific time intervals; the expected output for each scene is a dictionary that maps strings representing these relative time intervals of speech to

Table 4: Validation results on different random seed and temperature setting of Omni-CharM.

| Temperature | Seed | StoryQA | | | |
| --- | --- | --- | --- | --- | --- |
| | | Character | Action | Plot | Overall |
| | 12345 | 74.17 | 76.26 | 79.73 | 75.99 |
| | 23456 | 73.93 | 75.91 | 80.57 | 75.93 |
| 0.3 | 34567 | 73.59 | 75.91 | 79.66 | 75.58 |
| | 45678 | 74.02 | 76.22 | 80.29 | 76.02 |
| | 56789 | 74.02 | 76.58 | 80.43 | 76.16 |
| | 1234 | 73.19 | 76.09 | 80.08 | 75.54 |
| | 2345 | 73.47 | 75.95 | 81.21 | 75.86 |
| 0.5 | 3456 | 73.32 | 76.13 | 79.44 | 75.48 |
| | 4567 | 73.59 | 75.42 | 79.44 | 75.38 |
| | 5678 | 73.47 | 75.77 | 79.37 | 75.42 |

the corresponding ground truth speaker names. This data was constructed by processing distinct video scenes, for each gathering pre-extracted image frames, cast list information, the scene's audio, and subtitles formatted without speaker labels. The ground truth speaker identifications were then generated by utilizing existing detailed diarization data from StoryTeller to identify all spoken utterances within each scene and their precise timings relative to the scene's start, which were subsequently mapped to the known speaker names for those intervals.

- **Active Speaker Localization.** For the active speaker localization task, model should identify a specific known speaker within a video scene that contains only their single, complete utterance, and then tracking their face by providing a sequence of bounding boxes for each frame in that scene; the expected output is a dictionary mapping the verified speaker's name to a list of their adjusted and normalized bounding boxes across all frames of the scene. This data was created by first selecting video scenes extracted by Grounded Sam Ren et al. (2024) that exclusively contained one complete utterance from a single speaker, using detailed diarization data to establish the ground truth speaker. For these selected scenes, pre-detected faces in each frame were then matched against a cast library created from reference images by calculating cosine similarity; if a face was confirmed as the ground truth speaker for that scene, its corresponding bounding box was collected, and this trajectory of bounding boxes was subsequently adjusted and normalized to form the final output.

- **Character Emotion Recognition.** Model should identify the current speaker in the clip and choose one emotion adjective from 'angry', 'disgusted', 'fearful', 'happy', 'neutral', 'other', 'sad', 'surprised', '<unk>' given a video clip with audio and subtitles without speaker, cast list(character face and names). The data was constructed by splitting the videos into scene frames first. Then filter out audio segments that are longer than four seconds, feed the speaker's audio into the Speech Emotion Recognition model emotion2vec_plus_large Ma et al. (2024) and keep the samples where difference between the highest and second-highest sentiment scores to identify emotions exceeding one as the ground truth.

# B  VALIDATION ON OMNI-CHARM

We evaluate the stability of our model under different random seeds and temperature settings on StoryQA. As shown in Table 4, the results demonstrate that model achieves a mean overall accuracy of 75.94 ($variance = 0.046$) at temperature 0.3, and 75.54($variance = 0.037$) at temperature 0.5. The small variance across seeds indicates that our model is highly stable with respect to initialization randomness. Moreover, while temperature 0.3 yields slightly higher mean accuracy, both settings remain consistent within a narrow margin ($std \approx 0.2$), illustrating the robustness of our approach to improve model's character-level movie understanding ability.

**Video:** 

**Cast List:**  Zhang Daichen  Jiang Feng  Hui Lan

Subtitles of the input video with time stamps:

0.00 - 7.56 : In other words, if I want to have my memory erased again, I would need to undergo another surgery within seventy-two hours.

8.00 - 26.76 : To be precise, it's within 72 hours starting from the first memory reload. And it's not about taking it out, it's about permanent deletion, currently, there is no technology capable of extracting memories twice. During this time, Mr. Jiang Feng, you may also want to think it over. If you really decide to undergo the surgery, please let us know in advance.

30.00 - 30.60 : **'Then I'll make a reservation first.'**

30.50 - 31.71 : Xiao Feng

37.00 - 41.30 : Can't we wait until all the memories are restored before we decide? ……

**Question: What is Jiang Feng's response to the suggestion of the boss about memory erasing surgery?**

A. He denies the suggestion outright      B. He decides to think about it

C. He makes a reservation for the surgery      D. He asks for more details about the process

**Gemini's Answer:**

<think>1. **Analyze the User's Request:** The user wants me to answer a multiple-choice question based on a provided video clip, character information, and subtitles. The question is about a specific character's (Jiang Feng's) reaction to a suggestion about a memory-erasing surgery.

2. **Identify Key Information:**
   * **Characters:** The main characters involved in the relevant part of the conversation are Jiang Feng and the person he's talking to (let's call him the 'boss' or 'doctor', who is Zhang Daichen based on the character list and visual cues). Hui Lan also interjects.
   * **Question:** "What is Jiang Feng's response to the suggestion of the boss about memory erasing surgery?"
   * **Relevant Subtitles:** I need to find the part where the boss makes a suggestion and Jiang Feng responds. * **8.00 - 26.76: The boss/doctor explains the procedure and says,** "...Mr. Jiang Feng, you may also want to think it over. If you really decide to undergo the surgery, please let us know in advance." This is the suggestion/offer. * **30.00 - 30.60:** Jiang Feng says, "'Then I'll make a reservation first.'" This is his direct response to the suggestion. * **30.50 - 31.71:** Hui Lan interrupts by calling his name, "Xiao Feng". * **37.00 - 41.30:** Hui Lan then pleads, "Can't we wait until all the memories are restored before we decide?" This confirms that Jiang Feng has made a decision that she disagrees with.

3. **Evaluate the Multiple-Choice Options against the Evidence:**
   **A. He denies the suggestion outright:** This is incorrect. He does the opposite; he wants to schedule the surgery. **B. He decides to think about it:** This is what the boss *suggested* he do ("you may also want to think it over"). However, Jiang Feng's response is not to agree to think it over. He immediately decides to book it. So, this is incorrect. **C. He makes a reservation for the surgery:** His line, "Then I'll make a reservation first," directly supports this option. He is initiating the process of booking the surgery. **D. He asks for more details about the process:** This is incorrect. He is the one who starts the conversation by summarizing his understanding of the process ("In other words..."). The boss then clarifies some details, but Jiang Feng does not ask for more information after that. His response is a decision, not a question.

4. **Conclusion:** Based on the subtitle at 30.00 - 30.60, Jiang Feng's direct response to the offer of surgery is to make a reservation. Therefore, option C is the correct answer. The subsequent parts of the video where Hui Lan tries to dissuade him and later visits him in a different location are context, but the core of the question is about his immediate response in that initial scene. His response was decisive and immediate: "Then I'll make a reservation first." </think><answer>C</answer>

**Our Model's Answer:**

<think>1. **Identify the goal:** The task is to determine Jiang Feng's response to the boss's suggestion about memory erasing surgery.

2. **Locate the relevant scene:** The boss's suggestion about memory erasing surgery occurs between 0:00 and 0:07.56 in the video.

3. **Analyze the dialogue:**
   * The boss says, "In other words, if I want to have my memory erased again, I would need to undergo another surgery within seventy-two hours." This is the boss's suggestion.
   * Jiang Feng responds with, "To be precise, it's within 72 hours starting from the first memory reload. And it's not about taking it out, it's about permanent deletion, currently, there is no technology capable of extracting memories twice. During this time, Mr. Jiang Feng, you may also want to think it over. If you really decide to undergo the surgery, please let us know in advance."

4. **Evaluate the options:**
   A. He denies the suggestion outright: Jiang Feng does not deny the suggestion. He acknowledges the details and the time frame. B. He decides to think about it: Jiang Feng does not explicitly say he will think about it. He suggests thinking it over but does not confirm his intention to do so. C. He makes a reservation for the surgery: Jiang Feng does not make a reservation. He suggests thinking it over and letting them know in advance if he decides to go through with it. D. He asks for more details about the process: Jiang Feng does not ask for more details. He acknowledges the process and the time frame. E. None of the above: This is not correct because option B is partially correct.

5. **Conclusion:** Jiang Feng doesn't outright deny the suggestion, but he does not confirm his intention to think about it either. He suggests thinking it over and letting them know in advance if he decides to go through with it. This aligns most closely with option B, "He decides to think about it."

</think><answer>B</answer>

Figure 6: Case Study on Chain-of-Thought Result Interpretation.

# C  COT CASE ANALYSIS

In this section, we display the comparision of our model's chain-of-thought prompting result and Gemini2.5-Pro's chain-of-thought prompting reuslt. As illustrated in Figure 6, while our model learns the overall pattern of chain-of-thought steps, it fails to achieve precise and accurate analysis in the reasoning steps 2. To this end, our model performs badly when inferred with chain-of-thought prompting. It indicates the lack of cross-modality chain-of-thought reasoning ability for current Omni-MLLMs in such character-level movie understanding scenarios.

