# OpenReview forum: "All-in-One: Boosting Basic Capabilities in one Omni-MLLM to Enhance Movie Understanding"
_ICLR.cc/2026/Conference — Submitted to ICLR 2026_

### Official Review · Reviewer_TgwS · 2025-10-25

**Soundness:** 2
**Presentation:** 3
**Contribution:** 2
**Rating:** 4
**Confidence:** 4

**Summary:**

This paper introduces Omni-CharM, an all-in-one Omni-MLLM designed to improve ID-aware movie understanding, especially in long-form video contexts involving characters with complex relationships. The model is developed through a two-stage training pipeline: 1) a Basic Boosting Stage that introduces 12 types of ID-related multimodal tasks, frame and shot description data, and chain-of-thought (CoT) data, and 2) a Task Adaptability Stage where the model is further adapted to StoryQA-style question answering. Experiments on StoryQA and Video-MME benchmarks demonstrate improvements over open-source baselines and the original Qwen2.5-Omni-7B backbone, with ablation studies showing contributions from each component.

**Strengths:**

1. Overall, this paper addresses an important challenge in long-form & character-centric movie understanding, where many existing MLLMs operate at short clip or generic video QA levels.
2. The model demonstrates notable improvements on StoryQA, particularly in character-related reasoning.

**Weaknesses:**

1. My major concern lies in the reported performance on general video benchmarks. According to the tech report of Qwen2.5-Omni, Qwen2.5-Omni-7B achieves 64.3 (w/o sub.) and 72.4 (w/ sub.) on Video-MME, which is largely higher than the reported 61.6 in this submission. The authors should provide strong justifications regarding the performance mismatch.
2. While the proposed 12 ID-related tasks are well described, the quality of some automatically generated annotations (e.g., emotion recognition, diarization) may significantly affect training. The paper acknowledges this limitation, but a quantitative noise analysis would strengthen the claims.
3. The claim that the model is "all-in-one" with built-in basic capabilities may be slightly overstated, since these capabilities do not emerge intrinsically, but result from explicit task-specific supervised fine-tuning. The writing could moderate terms like "built-in" to avoid confusion and over-stating.

**Questions:**

Please refer to the weakness section for my questions.

---

### Official Review · Reviewer_pdST · 2025-10-30

**Soundness:** 2
**Presentation:** 2
**Contribution:** 2
**Rating:** 2
**Confidence:** 3

**Summary:**

The paper presents an instruction-tuning method for movie understanding. The assumption is that improving the character-ID information of existing MLLMs will improve their performance on movie tasks. Therefore, the proposed Omni-MLLM is first fine-tuned on a series of ID-related tasks (Basic Boosting Stage) and then adapted to the tasks to solve (Task Adaptability Stage). The method improves over the baseline Qwen2.5-Omni 7B.

**Strengths:**

- The paper presents interesting CoT experiments that confirm that CoT is not helping but show that it helps at training time.

- The results over the baseline are good.

**Weaknesses:**

Although the intuition behind character-ID information is interesting, there are several weak issues in this manuscript.
1/ Clarity.
The overall clarity of the work, both in terms of language and writing but more importantly in terms of messages and findings is very hard. I find the paper hard to follow and every part somehow disconnected from the overall goal.

2/ Lack of solid contributions.
The paper reads as if the main contribution is the Omni-CharM model, which is essentially a fine-tuned version of Qwen2.5-Omni 7B on character-ID information. Given that there exist already works that promote the intuition that character information is important (such as StoryTeller and IDA-VLM from Table 2) together with the fact that post-training is already well-adopted in the community makes the contribution of the work incremental and marginal.

3/ Generated data.
The proposed dataset is generated by relying heavily on very large models (Gemini 2.5-Pro and Qwen2.5-VL-72B), which risks reproducibility and scalability. Nonetheless, the analysis of the dataset is very low and we cannot understand the impact of each of these tasks; the various design choices for each task, albeit explained, are not experimentally verified and seem arbitrary.

4/ Lack of insights.
The method is presented as if having more data and ID information helps movie understanding but without more insights into what exactly is that existing MLLMs are missing.

5/ Evaluation protocol.
Omni-MLLM is evaluated on multiple-choice QA but open-ended or generative movie understanding tasks are not explored.

6/ Missing computational efficiency report.
Table 2 shows that the proposed method outperforms the baseline Qwen2.5-Omni 7B but without revealing the additional computational cost. I think this is an essential element missing into appreciating the impact, both during training and inference. This seems important as even within the paper we can see baselines that perform almost as well as the proposed model but require less resources: For instance, in Table 3 simply the F&S Desc (third row) performs almost as well as the full model with much less training compute and in Figure 3 using only character emotion recognition (less annotations, less data, less compute) performs also very well.

7/ Not verified generalization.
The proposed two-stage training is applied on top of Qwen2.5-Omni 7B. It would be interesting to examine whether fine-tuning other MLLMs could also improve performances.

8/ Data or structure or fine-tuning attribution.
It is unclear whether the improvement comes from the fact that (i) the model has seen more data; (ii) from the fact that the data are structured to include character-ID information or (iii) from the proposed fine-tuning setup.

9/ The paper is not standalone.
Several parts are not clear and we are pointed to external papers to understand; even for core parts of the work, such as the second part of the method in Section 3.4.

**Questions:**

For questions, it would be great if the authors could address some points from the weaknesses above. Specifically:

Q1 (W2, W4) Clarify contributions and insights.

Q2/ (W6) Compute.

Q3 (W3) Perhaps examine how data quantity and data generation with open small models affect the performance?

Q4 (W5) Examine performance on other evaluation protocols (open-ended).

Q5 It would be great if the authors could provide some failure case analysis.

Q6 Perhaps discuss in terms of ethics the gender bias, and representation and potential misuse of character-ID.

Q7 (W8) This would be an interesting experiment to have but, of course, there will be no time to examine this; however, having an intuition of where the improvement(s) comes from would be very useful.

---

### Official Review · Reviewer_wY84 · 2025-11-01

**Soundness:** 3
**Presentation:** 2
**Contribution:** 3
**Rating:** 4
**Confidence:** 4

**Summary:**

This paper poposes Omni-MLLM for movie understanding, which can combine visual, textual, and audio information. They construct identity related data to enhance capability of ID identification, and leverage frame and shot descriptions to improve model's performance. Finally, they explore the influence of CoT in movie understanding. They achieve best performance on StoryQA.

**Strengths:**

1. Movie understanding is a significant problem for MLLM. This paper proposes Omni-MLLM to combine all modalities to understand movies, which is essential.
2. The constructed ID-related data is comprehensive and diverse. The exploration of CoT is interesting. Because the learning of CoT improves model's understanding ability, so the metrics get higher, but if inference with CoT, it may bring some errors in the intermediate process. Improving CoT's accuracy in inference is a challenge.

**Weaknesses:**

1. The writing need be improved. The abstract and introduction should be organized better. For example, 'alleviate the difficulty of training.' in the abstract, and line 78, 'other models that focus on distractors' is not clear.
2. Task Adaptability Stage in the dual-stage should be more clear. In this version, readers don't know how it is conducted, and its effectiveness in the experiments. Compared to the first stage, the second stage is too less detailed, maybe you need change the name of 'dual-stage'.
3. The construction of ID-related data need more explanation in the main text. Because the data construction is this paper's main contribution.

**Questions:**

1. See weakness.
2. List differences between Storyteller and Omni-MLLM.

**Details Of Ethics Concerns:**

The copyright of movie data.

---

### Meta-Review · Area_Chair_rNSB · 2025-12-21

**Summary:**

The paper proposes Omni-MLLM, an approach that improves movie understanding by integrating visual, textual, and audio cues with enhanced character identity (ID) information. Through a two-stage training pipeline—with a Basic Boosting Stage introducing a diverse set of 12 ID-related multimodal tasks and a Task Adaptability Stage focusing on movie-related question answering—the authors aim to boost performance on tasks such as StoryQA and Video-MME. While the submission shows promise by reporting improvements over the Qwen2.5-Omni-7B baseline and by providing interesting chain-of-thought (CoT) experiments, concerns remain about the clarity of presentation and the reproducibility of the proposed framework.

**Reviewer Concerns:**

All reviewers highlight issues with both clarity and experimental rigor. The manuscript suffers from writing and structural inconsistencies, ranging from an inadequately organized abstract and introduction to significant gaps in the detailed explanation of the Task Adaptability Stage and the ID-related data construction. Several reviewers expressed concerns about the reliance on automatically generated annotations, which raise questions about reproducibility, scalability, and the actual contribution of the proposed fine-tuning to character-ID information. In addition, there are doubts regarding the computational efficiency of the proposed method and whether its performance gains stem primarily from additional data or from the structured training approaches. Questions regarding the evaluation of generative and open-ended tasks further compound these concerns.

**Reviewer Scores:**

While aspects of the work show potential, the reviewers' aggregated feedback indicates that the paper does not yet meet the required standards for acceptance. The incremental contributions, coupled with an unclear methodology and insufficient experimental analysis, raise significant doubts about both the reproducibility and the overall impact of the work. Therefore, the meta recommendation is to reject this submission in its current form, with a suggestion to address the outlined concerns in a future revision.

---

### Decision · Program_Chairs · 2026-01-26

Reject